# Tipsters and Addiction in Spain. Young People’s Perception of Influencers on Online Sports Gambling

**DOI:** 10.3390/ijerph18116152

**Published:** 2021-06-07

**Authors:** Juan Enrique Gonzálvez-Vallés, José Daniel Barquero-Cabrero, David Caldevilla-Domínguez, Almudena Barrientos-Báez

**Affiliations:** 1Communication Theories and Analysis, Faculty of Information Science, Complutense University of Madrid, 28040 Madrid, Spain; davidcaldevilla@ccinf.ucm.es; 2Marketing Department, ESERP Business & Law School, 28016 Madrid, Spain; jd.barquero@eserp.com; 3Education Department, European University of Madrid, 28108 Madrid, Spain; almudena.barrientosbaez@unir.net

**Keywords:** tipsters, addiction, gambling, young people, regulation

## Abstract

This research analyzes young people’s perception of the presence of tipsters as influencers on online sports gambling and whether their presence can promote addiction to this activity. To achieve this goal, we designed a questionnaire that was administered to young people in public universities in Madrid, being answered by 1032 individuals, out of whom 613 claimed to be regular bettors. We proceeded to the factor analysis of the variables with a high or very high correlation, and results showed that young people perceive a clear relationship between gambling and addiction. An even more enlightening aspect is the result that links tipsters with addiction to online sports gambling; young people’s perception correlates both concepts with extraordinary strength. This study’s main conclusion makes it clear that there is a huge amount of influence of tipsters on the world of online sports betting, as well as the risk of marrying these two concepts, since young people perceive that either they or others could be initiated into the world of problem gambling.

## 1. Introduction

### 1.1. Young People and Gambling

Previous investigations have not clarified what demographic and psychological factors define the profile of online gamblers regarding sporting events. There are studies that affirm that the higher a person’s cultural and educational background is, the less likely they are to acquire gambling disorder since their emotional intelligence has been further developed [1,2]. However, other researchers in this field claim that it is common for people holding high office to exhibit addictive tendencies towards gambling due to their socioeconomic status, which allows them to risk huge quantities on leisure. As for the age of people who suffer from gambling addiction, the most predominant ages range from 17 to 24 years old, and the reason is that this sector is particularly vulnerable to the harms related to online formats due to its generational characteristics [3], completely immersed in the online world and connectivity. Studies demonstrate that these people pay special attention to advertisements for sports betting, since they connect with a sector of entertainment especially popular among the youth [4].

The fact of individuals surrounding themselves with online bettors promotes their introduction into the world of online gambling [5]. Secondary (friends) and tertiary (people who we admire) reference groups particularly affect the decisions young people make because they are going through their maturing stage [6,7]. In addition, this fact entails normalizing certain behaviors that would not be perceived as normal in a different environment. Thus, the dividing line of what the person had initially understood as wrong begins to blur [8].

Another behavior that indicates gambling addiction and that promotes hazardous practices on the Internet is betting on unregulated websites and/or having several accounts on different betting portals. It is a clear indicator of the psychological harm caused by people who support risky experiences and impulsive behaviors [9].

Finally, social networks are used by young people, making them the perfect medium to generate the engagement that online bookmakers vigorously seek [10]. Due to their lack of knowledge, innocence, and self-confidence, it is the youth who are the most interested and active in online betting, which is noted in their activity on social networks, especially by following profiles that reflect similar lifestyles to theirs [11,12]. These circumstances promote anytime–anywhere betting, which, together with it being an activity that can be performed in total privacy, does not bear a socially negative connotation. All this leads to the acquisition of addictive habits towards gambling [13].

### 1.2. Online Gambling in Spain

Online gambling is the activity by which the user wagers and performs actions related to betting, offered on platforms that can be accessed via the Internet. It is common to gamble through multiple devices connected to the Web, such as computers, mobile phones, or tablets. In many cases, these devices are interconnected and thus further facilitate user’s accessibility to these platforms. Hence, the worldwide number of online gamblers is considerably higher than the number of gamblers in traditional bookmakers [14].

According to data from the Ministry of Health [15], online gambling is the second cause of problem gambling, behind slot machines, and it is an addictive product. Sports betting accounted for 40% of the EUR 17 billion that are annually wagered on the Internet in Spain, according to the Ministry of Finance. In contrast to slots, online gamblers feel that, based on their knowledge of gambling and their at-half-time-of-the-match decisions, they have everything under control. In the mind of the user, this activity is not associated with luck, let alone with problem gambling at first [16]. Here the figure of tipster comes into play, which is going to be defined further on in the text.

This is because it is a more impulsive and automatic action, within reach anywhere and anytime. There is no need to move around, to adapt to opening hours, and no peer pressure or a negative response from the social environment [17] due to the excessive amount of time spent on these betting sites. In other words, online gambling is favored by the number of adherents it has already accumulated, given its private and omnichannel nature, and improvements in connections to the Internet all around the world, as well as the notifications of the updates provided by the very betting applications, which act as an advocacy group 2.0 [18,19].

Excessive use of online gambling could contribute to developing gambling problems. It is estimated that, in Spain, disordered gambling affects 9% of the population, according to the Spanish Government [20]. Gambling addiction is considered the first addiction related to human behavior, which could serve as a clue or precedent for other diseases associated with excess and/or impulsiveness. Psychological studies regarding this subject affirm that as the Internet has been progressively evolving, and in turn, online bookmakers, significant trends have developed, which indicate the nonstop annual increase in gambling addiction [21,22].

In Spain, the Directorate General for the Regulation of Gambling (DGOJ), appointed under the Ministry of Finance, is responsible for regulating, authorizing, supervising, controlling, and sanctioning gambling. These data are found in its last report [23]:In Spain, online gambling accounted for EUR 67.54 million in March 2020.In 2019, this sector’s expenditure on advertising and marketing grew by 6.5% and 42.8% respectively.Out of the EUR 18,778.86 million turned over on online gambling in 2019, gambling companies invested 180 million in advertising.The number of online gamblers in Spain went from 289,770 in 2013 to 937,783 in 2019.Among all the forms of online gambling, the preferred choice is live sports betting, which accounted for EUR 384.17 million in 2019.The mean number of active gamblers was 909,513 in the first three months of 2019 alone.Bookmakers have quadrupled their bond investment over the last five years: it was EUR 30.5 million in 2013 and EUR 123.2 million in 2020.

### 1.3. Gambling Influencers: Tipsters

Online gambling is growing rapidly in terms of popularity due to the constant boom in products and, more importantly, the increase in market and portfolio shares. According to the study conducted by the Center for Gambling Education and Research of the Southern Cross University, the online gambling market was valued at EUR 6.1 billion in 2013, with expected annual growth of 10.1% in 2018 [14].

The figure of “tipster” emerges in this expansion context: “A tipster is a forecaster in gambling who analyzes different sporting events, and identifies odds with value in them. Generally, they tend to be winning bettors over the long term, whose statistical estimations are the seal of approval seen by the public” [13].

The business model is two-fold: on the one hand, they make money betting on their own, and, on the other hand, they obtain extra income by sharing their tips with subscribers, who have previously paid to have access to such tips. In addition, they simultaneously charge bookmakers for bringing new gamblers or for offering forecasts, which may or may not happen. These tipsters proliferate without control or regulation, especially on social networks, which have become an important asset that gambling companies utilize as a marketing tool to increase their share of users [24].

Tipsters must have two characteristics that cause users to choose one in particular: profitability and rapidity. They surround themselves with celebrities and show a luxurious life while implying that all those comforts derive from online gambling. Some of them have a turnover that ranges from EUR 20 to 30 thousand per month. The media interview them as experts or victors who have managed to make a living out of their bets. Some channels have over 100 thousand followers. All that fame and success cause other people to imitate their behavior aspiring to be and live like them [25,26].

Tipsters tend to specialize in a specific sport or type of competition. Their results are unequal in the different sports they analyze; they even have a high percentage of errors, which they try to hide by distorting their statistical estimations. In Spain, according to DGOJ (Directorate General for the Regulation of Gambling), it is estimated that approximately 50 thousand gamblers regularly follow their recommendations, but there could be many more [27].

Despite there being thousands of gamblers who claim to have been scammed, almost none of them report it. It is not clear either whether tipsters are doing something illegal or if they are just acting as socializing agents, like the style of what the print media did in the past [28], although during times of post-truth and communication 2.0 [29]. In addition, these sums are usually small, and nobody wants to acknowledge that they have been scammed [27]. An experiment conducted for four months [30] consisted of monitoring the activity of the Spanish tipsters with the highest number of followers, both on their free and paid channels, on which, according to them, they share their best forecasts with higher accuracy rates. This is the main reason why online gamblers are willing to pay monthly fees. Additionally, 15 of the most successful tipsters in Spain were interviewed. This research concluded that these forecasters’ activity is very similar to that of televangelists or tarot fortune tellers: practically, gamblers blindly trust in them. These forecasts do not allow making a living from betting, since they actually pretend to have made a lot of money from online gambling, although most of the time they do not even place the bets, let alone wager such high sums.

The figure of the tipster comes from a simple reality: the figure of an expert [31]. However, the experts who might be analysts and forecasters did not work as bookmakers. Hence, they charge fees for taking the risk of receiving the money of gamblers who hire their services.

In Spain, the demand for analyses, forecasts, or picks offered on communities, portals, forums, and blogs has been known for a while. The turning point was the massive participation on generalist forums such as ForoCoches, on which discussions about gambling took place and picks (choices that forecasters select among all the supply options of sporting events) were published. This was something that attracted gambling enthusiasts who then began to ask for advice from those experts. It was at this point that the figure of the tipster emerged [32].

Since the boom of social networks, and with them the emergence of influencers being paid by companies to publicize themselves in practically all the consumption sectors, this new profession has become the most relevant trend in the sports gambling industry in Spain [33].

Tipsters tend to provide their forecasts—paid or free—via different means. Betting sites and blogs and social networks (e.g., Facebook and Twitter) are tipsters’ typical free channels. For paid channels, they tend to use platforms with limited access, such as e-mails, WhatsApp, Telegram, or websites’ inner pages (which are platforms that can only be accessed with previous identification by signing in) [34].

In Spain, the sports on which tipsters focus are football, basketball, and tennis. Their success is because of their high mediatization on television and the radio, hence causing their target audience to be larger [35,36,37].

Football: Tipsters tune in for the best sporting events: La Liga, the Champions League, the UEFA Champions League, etc.Basketball (NBA): Tipsters also analyze basketball matches, focusing on the Liga ABC, the European competitions of clubs, and the American NBA. European and global competitions are analyzed, both women’s and men’s.Tennis: To tipsters, tennis is one of the sports through which their labor can be more easily monetized, since a large number of matches take place throughout the year.

#### 1.3.1. Legal Framework

Currently, the online gambling business is going through an expansion period, but it lacks real and total control from the Administration. According to data from DGOJ, during 2018, around 1.47 million people gambled online in Spain (around 3% of the population), of which 300,000 were new to this activity. Out of them, sports betting attracted 48% of all gamblers: this percentage increases every year, with an exorbitant profit for bookmakers [23].

Gambling’s lobby, with the traditional use of their influence [38], affirm that their sector is pretty much regulated, however, Telegram tipsters entail a handicap that nobody controls: neither how they advertise themselves, nor the statistical estimations that prove their knowledge and profits, nor their incitement to gambling. Companies have been created exclusively to be intermediaries between tipsters and gamblers. These types of applications allow tipsters to be supported by an external company that verifies their forecasts in an apparently official way. On another note, the app offers tipsters the possibility of hiding their statistical estimations or at least part of them if they are not accurate enough.

The minister of Consumer Affairs, Alberto Garzón, presented the Royal Decree Law 11/2020, of 31 March, of the Ministry, to regulate both online and offline gambling in Spain, aiming to improve the current Law 13/2011, of 27 May. Regarding tipsters and their social networks, it states: “tipsters, gambling influencers, shall be obliged to show their real data on profits and accurate tips rates to avoid the deception that winning is easier than what it actually is” [39]. In conclusion, the figure of tipsters is not only related to their abilities as forecasters; they themselves also become marketing products that, ultimately, ought to be regulated [30].

#### 1.3.2. Aim and Objective

The main objective of this research is to know young people’s perception of the figure of tipsters and whether they think tipsters are generators of addictive behaviors, in this case, of gambling.

This general objective brings us to the following research questions (RQs):RQ 1: What is the level of presence of online sports gambling in the life of young people?RQ 2: What is tipsters’ level of knowledge and their influence on the bets being placed?RQ 3: Is there a relationship between tipsters and addiction to online sports gambling?

## 2. Materials and Methods

We conducted a preliminary descriptive analysis based on the premise of a generalized linear regression model with logical functions linked to it. For analysis and collection of data, we created a survey with questions based on previous investigations [40], whose construct and content validity were assessed by experts [41]. The questions were structured through a 5-to-7-item Likert scale, and, subsequently, the most relevant variables were selected, some of which were standardized and grouped for exploratory factor analysis. In any case, reliability between variables was measured using Cohen’s Kappa coefficient, resulting in K = 1 in all the variables.

To establish our research we created an 18-item online questionnaire that was distributed from 10 October 2020 to 20 March 2021 among the students of public universities in Madrid, being the universe comprised of 196,823 people, hence generating an effective sample (ES) of 1032 individuals, with a 95% confidence level and a margin of error of 3.04%. All ethical aspects were taken into account when conducting this research. The selection of people was carried out in the different faculties by distributing a QR code with which the study could be accessed. Informed consent was presented at the beginning of the survey conducted through Google Forms to preserve the anonymity of the sample.

The minimum age of the sample collected was 17 years old and the mean age was 18.9 years of age. The gender distribution was 57% women (n = 588) and 43% men (n = 444). The quantitative research stage entails a series of steps that involve explaining in detail the statistical analysis that determines the methodology used in this investigation. IBM SPSS v.25 software (IBM Corp. Released 2017. IBM SPSS Statistics for Windows, Version 25.0. Armonk, NY: IBM Corp.) program was used to analyze the data collected. After cleaning the variables, we explored the correlations between variables to know if they shared a latent structure. After the variables were standardized, we proceeded to the reduction of factors and to implement the KMO test to establish the suitability for factor analysis, evaluating the existing correlation between the variables and the existence of a latent structure between them. Similarly, Bartlett’s test of sphericity was used to analyze the applicability of factor analysis to the chosen variables. Cronbach’s alpha was implemented to assure the verification of the research tool’s effectiveness [42].

Subsequently, Pearson and Spearman’s methods were implemented to verify if there were links between the factors analyzed.

## 3. Results

The analysis of the results links the variables that articulated the research questions with all the aspects related to online gambling, the presence of tipsters, and their influence. Regarding RQ1, the presence of online sports gambling in the life of young people, the total corpus was distributed in 59.4% (613) who had gambled on online sports betting sites in the past 12 months and 40.6% (419) who had not done so. To complete the analysis of the remaining items in the questionnaire, we proceeded to filter the sample to obtain only the data of the active users in online sports gambling. To address RQ1 in a comprehensive manner, we proceeded to analyze the correlations between this first question and the other two related aspects, namely the frequency of bets being placed and the amount of money available (See Table 1).

The correlations observed above show that the questions inquiring about the presence of online sports gambling in the life of young people have a *p*-value lower than 0.05, which means they are statistically significant. All the correlations are positive, and the correlation with the lowest value (0.632) has a very strong relationship between “participation in online gambling” and “amount of money available”. The highest value (0.830) also shows a very strong relationship between the frequency of bets being placed and the amount of money available.

After studying the variables regarding RQ1, all those variables alluding to RQ2 were analyzed, meaning tipsters’ level of knowledge and their influence on the bets that young people place. Just like the previous table, we proceeded to study their correlations by applying Pearson’s correlation coefficient (See Table 2).

Again, the level of significance indicated that the correlation between variables was significant and positive. The lowest value in this new analysis (0.711) already shows a very strong correlation between following tipsters and the bets being placed due to the tips given by these types of influencers. The highest result (0.917) shows a very strong correlation between the bets being placed due to the tips given by tipsters and the fact of more money being wagered after receiving their recommendations.

The last step in this part of our research established some final correlations between those variables concerning RQ3, that is, the relationship between tipsters and addiction to online sports gambling (See Table 3).

In this case, the results show that the correlations are significant and positive. The correlation is very strong only in the relationship between the knowledge of friends being addicted to online sports gambling and tipsters’ influence on them (0.818). However, the correlation is weak or moderate in the remaining correlations, and there is only one strong correlation left (0.374), the relationship between subjects’ perception of their own addiction to online sports gambling and the influence that tipsters exert on their bets.

Subsequently, we conducted the KMO test, and, since it was very close to 1, the Kaiser–Meyer–Olkin measure of sampling adequacy indicated a strong relationship between the variables. Similarly, Bartlett’s test of sphericity was conducted, resulting in a value lower than 0.05, thus indicating the suitability of data for exploratory factor analysis (EFA) (See Table 4).

The total explained variance, grouped into two factors, only explained 43.75% of the variance, while being grouped into three factors raised it to 77.8%. The extraction of the rotated component matrix agglutinated the variables in a similar way as they had been observed in the high and very high correlations (See Table 5).

With the Cronbach’s alpha test being standardized, we verified that the correlations between the constructs were reliable. Since they are higher than 0.600, their validation can be asserted. The factors were labeled as Gambling, Tipsters, and Addiction (See Table 6).

After standardizing the variables and grouping them into the three factors, we chose and specifically calculated the Person’s correlation coefficient and Spearman’s correlation coefficient, with a significance level set to 0.05 (See Table 7).

After observing the table, we can note that the strongest existing relationships between variables correspond to those concerning tipsters, both regarding their influence when it comes to placing bets and their role when it comes to this habit becoming an addiction (r = 0.731). The correlation is very strong and positive, meaning that the higher the influence of tipsters on gambling is, the more they are regarded to have an influence on creating addicts.

There are also strong and positive correlations between young people who gamble and the influence that tipsters exert on them (r = 0.643). The last aspect studied in this analysis also relates gambling to addiction (r = 0.602). After confirming all these aspects through Pearson’s Correlation, we conducted a confirmatory analysis of the study through Spearman’s Correlation (See Table 8).

It can be observed that the ranks between variables underpin the ones previously obtained through Pearson’s correlation. In fact, the correlation between tipsters and addiction (rho = 0.722) and the correlation between gambling and addiction (rho = 0.705) confirm that there is a relationship between them, just as aforementioned. Similarly, the strong correlation between the variables gambling and tipsters can be noted, with a *p*-value equal to absolute zero, and rho = 0.703.

## 4. Discussion

This research relates the figure of influencers on online sports betting, tipsters, to the action of placing bets and the possible risk of transitioning to an addiction such as problem gambling [2,7,11,12]. The objective is to know young people’s perception of the figure of tipsters and whether they think tipsters are generators of addictive behaviors, in this case, of gambling [10,13]. Gambling is legal, but its abuse may lead to a nonsubstance addiction, or even a disorder, as claimed by the World Health Organization [43].

The results of the analysis show that gambling and the presence of tipsters are worthy of a thorough reflection. It is true that the world of gambling has been subjected to significant restrictions in its more traditional facet, such as casinos and slot machines. In fact, access was denied to minors in the case of places dedicated to gambling, and their presence in the media was practically inexistent [7,13]. Nevertheless, the appearance of online sports betting has entailed the rupture of that model, and advertisement for that mode of gambling has increased exponentially. It also led to the emergence of tipsters, as an example of persuasion on social networks, taking advantage of the fact that there are no specific laws to regulate commercial communication, neither in Spain nor in Europe [10,19].

However, the appearance of addictive behaviors related to these types of gambling among young people has caused important social consequences. This has promoted, in many countries, the initiation of regulatory processes, such as in Spain [31]. The law of the Ministry of Consumer Affairs addresses especially the figure of tipsters and their modus operandi. It explicitly limits their actions, both in how they incite gambling and the accuracy of the data they provide.

The analysis of the three factors has shown high correlations, especially in the case of tipsters and addiction to online sports gambling. The importance of these data lies in the fact that these types of influencers hold, in their communications, enough attraction to cause young people to believe that they can lead someone (themselves or other people) to addictive behaviors [11]. There are also important ethical considerations for tipsters regarding the psychological manipulation of people, leading them to pernicious habits [19].

As for this study’s central subject, which analyzes young people’s perception of tipsters, the results reveal that there is a clear and positive (in statistical terms) relationship between gambling, tipsters, and addiction. This suggests that more bets are placed when tipsters intermediate these processes, and this can be explained according to Cialdini [44] and the theory of decision making. It is worth considering that the more people follow their tips, the more probabilities there are of developing addiction. The high impact of tipsters on the bets that young people place and the fact of more bets being placed more frequently, with more money involved, cause young people to perceive that anyone could transition more easily to problem gambling [18].

This research aimed to know young people’s perception of tipsters and whether these influencers could lead them to addiction, but after analyzing the results, we concur with the postulates of Perelló, Muela, and Romero [45] regarding the particularly vulnerable publics in this area. The characteristics of the development period of the group of respondents allow us to refer to them as emerging adulthood [46], and they are exposed to unregulated messages that could fit the concept of fake news, and yet, there is no dissuasive or punitive regulation concerning this aspect. Addiction to online sports gambling must be considered a public health issue, and it should not be addressed from the perspective of consumption alone. Similarly, other vulnerable publics, such as children, are exposed to these risks through the indiscriminate use of smartphones. In fact, the risk is not only limited to problem gambling since other disorders could be involved, in a direct way or as a consequence of gambling addiction [47].

### 4.1. Implications

Previous works have already warned about tipsters’ behaviors and their influence capacity [46]. Even their predictive capability has been studied, measuring its reliability and its connection with the flow of gambling markets [47,48]. In fact, there are also publications about the influence of tipsters on diagnosed problem gamblers [49].

However, this research makes clear that the marriage of young people, gambling, and tipsters tends to lead to addiction, something that even the subjects of this study perceive. The Public Administration must consider banning the use of these types of influencers in the communication of online sports gambling. The limitations must extend to the level of other products, such as drugs, and glance over other fields, in which the use of influencers is generalized, such as the fashion industry [50].

Bearing in mind that many of the bookmakers that use tipsters are not established in Spain, the legislation should not be limited only to the Spanish ambit, but it should extend to the European context. The European regulations must homogenize the national legislations to walk over one single path that prevents any type of technological constraint. Finally, it should be noted that scholars of the causes and consequences of addictions highlight the necessity of educating the public about personal control, resilience management, and emotional intelligence from an early age, interweaving educational curricular content and this cross-disciplinary content [51], and synchronizing the new pedagogy to these goals [52,53,54].

### 4.2. Limitations and Future Research

This study’s main limitation is that it is focused on demonstrating the correlation between gambling, tipsters, and addiction, while future works should delve into noteworthy variables such as segmentation by gender, age, or financial capacity. Another limitation is that the analysis does not distinguish what types of products tend to lead more to problem gambling (e.g., welcome bonus, multiple bet) or the moment of placing a bet (i.e., before the sporting event occurs or while it is taking place). Similarly, the collective or individual nature of bets or their amount will be the subject of study of future works.

The main focus of this research is young university students, but future investigations could widen the age band, both upwards and downwards, to cover segments completely, such as millennials, centennials, or Generation Z. Similarly, other Spanish regions could provide a more holistic vision of the country’s situation, and even an analysis at a European level can be conducted.

## 5. Conclusions

The use of tipsters in online sports gambling is experiencing an unprecedented boom, taking advantage of the current legal vacuum. The influence of these types of actors is perceived, by young people, as a factor that promotes addiction to these kinds of behaviors. The existence of tipsters is also justified from a perspective of supporting the placing of more bets and larger sums of money at stake.

Young people clearly perceive that the use of tipsters leads to these types of habits, which, either in them or other people, generate problem gambling. The trust in tipsters’ messages as well as the communication of data regarding gambling cause people to place sports bets online. Similarly, young people associate these factors with addiction as a direct consequence.

## Figures and Tables

**Table 1 ijerph-18-06152-t001:** Correlation table regarding RQ1.

Correlations		Participation in Online Gambling	Frequency of Bets Being Placed	Amount of Money Available
Participation in online gambling	Pearson C.	1	0.691	0.632
	*p*-value		<0.001	<0.001
	N	613	613	613
Frequency of bets being placed	Pearson C.	0.691	1	0.830
	*p*-value	<0.001		<0.000
	N	613	613	613
Amount of money available	Pearson C.	0.632	0.830	1
	*p*-value	<0.001	<0.001	
	N	613	613	613

**Table 2 ijerph-18-06152-t002:** Correlation table regarding RQ2.

Correlations		Following Tipsters	Number of Tipsters Followed	Trust in Tipsters	Bets Placed Due to Tips Given by Tipsters	Increase in Bets Due to Tips Given by Tipsters
Following tipsters	Pearson C.	1	0.717	0.755	0.711	0.75
	*p*-value		<0.001	<0.001		
	N	613	613	613		
Number of tipsters followed	Pearson C.	0.717	1	0.867	0.899	0.775
	*p*-value	<0.001		<0.001	<0.001	<0.001
	N	613	613	613	613	613
Trust in tipsters	Pearson C.	0.755	0.867	1	0.904	0.683
	*p*-value	<0.001	<0.001		<0.001	<0.001
	N	613	613	613	613	613
Bets placed due to tips given by tipsters	Pearson C.	0.711	0.899	0.904	1	0.917
	*p*-value	<0.001	<0.001	<0.001		<0.001
	N	613	613	613	613	613
Increase in bets due to tips given by tipsters	Pearson C.	0.75	0.775	0.683	0.917	1
	*p*-value	<0.001	<0.001	<0.001	<0.001	
	N	613	613	613	613	613

**Table 3 ijerph-18-06152-t003:** Correlation table regarding RQ3.

Correlations		Addiction to Online Sports Gambling	Friends with Addiction	Tipsters’ Influence on Me Concerning Gambling Addiction	Tipsters’ Influence on Others Concerning Gambling Addiction
Addiction to online sports gambling	Pearson C.	1	0.135	0.374	0.214
	*p*-value		<0.001	<0.001	
	N	613	613	613	
Friends with addiction	Pearson C.	0.135	1	0.183	0.818
	*p*-value	<0.001		<.001	<0.001
	N	613	613	613	613
Tipsters’ influence on me concerning gambling addiction	Pearson C.	0.374	0.183	1	0.147
	*p*-value	<0.001	<0.001		<0.001
	N	613	613	613	613
Tipsters’ influence on others concerning gambling addiction	Pearson C.	0.214	0.818	0.147	1
	*p*-value	<0.001	<0.001	<0.001	
	N	613	613	613	613

**Table 4 ijerph-18-06152-t004:** KMO and Bartlett’s test.

Kaiser–Meyer–Olkin Measure of Sampling Adequacy		0.934
Bartlett’s Test of Sphericity	Approx. Chi-square	9.758113
	Df	80
	Sig.	<0.001

**Table 5 ijerph-18-06152-t005:** Rotated component matrix.

	Component 1	Component 2	Component 3
Zscore. Participation in online gambling	0.754		
Zscore. Frequency of bets being placed	0.701		
Zscore. Amount of money available	0.698		
Zscore. Following tipsters		0.739	
Zscore. Number of tipsters followed		0.733	
Zscore. Trust in tipsters		0.712	
Zscore. Bets placed due to tips given by tipsters		0.701	
Zscore. Increase in bets due to tips given by tipsters		0.697	
Zscore. Addiction to online sports gambling			0.855
Zscore. Friends with addiction			0.832
Zscore. Tipsters’ influence on me and gambling addiction			0.821
Zscore. Tipsters’ influence on others and gambling addiction			0.818

**Table 6 ijerph-18-06152-t006:** Cronbach’s alpha.

Variables	Items	Cronbach’s Alpha
Gambling	3	0.883
Tipsters	5	0.845
Addiction	4	0.921

**Table 7 ijerph-18-06152-t007:** Pearson’s correlation between the factors.

Correlations		Gambling	Tipsters	Addiction
Gambling	Pearson C.	1	0.643	0.602
	*p*-value		<0.001	<0.001
	N	613	613	613
Tipsters	Pearson C.	0.643	1	0.731
	*p*-value	<0.001	<0.001	
	N	613	613	613
Addiction	Pearson C.	0.602	0.731	1
	*p*-value	<0.001	<0.001	
	N	613	613	613

**Table 8 ijerph-18-06152-t008:** Spearman’s correlations between the factors.

Correlations		Gambling	Tipsters	Addiction
Gambling	Correlation coefficient	1	0.703	0.705
	*p*-value		<0.001	<0.001
	N	613	613	613
Tipsters	Correlation coefficient	0.703	1	0.722
	*p*-value	<0.001	<0.001	
	N	613	613	613
Addiction	Correlation coefficient	0.705	0.722	1
	*p*-value	<0.001	<0.001	
	N	613	613	613

## Data Availability

Data sharing not applicable.

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
