# Peer review of "Tipsters and Addiction in Spain. Young People’s Perception of Influencers on Online Sports Gambling"

_ijerph, 2021, doi:10.3390/ijerph18116152_

Round 1

Reviewer 1 Report

This is a controversial and highly topical issue which should be studied and studied in greater depth because of its potential impact on young people.

The field study is very successful, since it gives us an x-ray of today’s society and allows us to define strategies to control the phenomenon, as well as to give guidelines for action.

At a technical level, the work seems to me to be well structured, with a clear methodology and real objectives that help to understand the article presented.

The findings are correct and open up interesting lines of research.

Author Response

Thank you so much for your contributions. It has been a pleasure to receive such a review. We have edited the English language and style required.

Reviewer 2 Report

The article presented for review: Tipsters and addiction in Spain. Young people's perception of influencers on online sports gambling

deals with the socially important issue of behavioral addictions on the example of addiction to gambling. The problem, unfortunately, concerns more and more people in different age groups. The article is a good supplement to the literature on the subject in terms of the analysis of the influence of people typing results in sports betting. Thus, the content of the article fits into the practical context of anti-gambling prevention. The article has been correctly formatted and divided into parts as required by the journal. It is an interesting scientific analysis of the factors contributing to the spread and strengthening of gambling inclinations among young people.

However, the entire article requires some, in my opinion, significant corrections before publication, which should be introduced to the text in the following scope and parts of the article

Introduction:

  • A separate part should be introduced in the body of the article, where the aim of the work, problems and research hypotheses will be presented. Currently, this entry is at the end of the Introduction. It is worth highlighting the main methodological part of the work in a separate section of the article.

Materials and Methods

  • It is worth describing in detail how the survey was conducted in terms of whether it was on-line or paper and pencil
  • The text does not contain a reference to the ethical context of research, including the method of selecting people for research, the method of obtaining consent from respondents to participate in research, the manner of preserving the anonymity of respondents, voluntary research and the possibility of withdrawing as well as the security of data archiving if they were obtained electronically through the replication of research questionnaires (e-mail, scientific research platform, etc.)
  • I would like to ask authors for a correction of the entry in the number of (n) groups of respondents: The gender distribution was 57% women (588) and 43% men (444). Correct: The gender distribution was 57% women (n = 588) and 43% men (n = 444).
  • It is worth providing examples of items that were in the questionnaire

Results

  • Tables 1, 2, 3, 4, 7, 8 - It is better to report the actual p-values ​​than just the levels below .05. If they are less than .001, then the less than <.001 can be used, but otherwise report all p-values, even if they are above 0.05. This is due to the reduction of reporting bias and if there are meta analyzes, the p-values ​​can be used. The notation of p-value = .000 is not correct

Discussion

  • I would suggest a greater reference than before to the research that was cited in the theoretical part of the article, but are not discussed in the context of own results
  • Conducting a comparative discussion of the obtained results with similar research on issues from other countries (if such were conducted)
  • The analysis of research conclusions should be deepened and directly stated. The influence of tipsters can be discussed in the context of techniques of influencing people according to Robert Cialdinii and the theory of decision making
  • It is worth briefly referring to the characteristics of the development period of the group of respondents called "emerging adulthood" (theory by J. Arnett) as the one where factors influencing the influence of tipsters on youth behavior can be searched for.

Author Response

Thank you very much for your contributions. You can observe the changes in the attached file
